# Polymer- and Hybrid-Based Biomaterials for Interstitial, Connective, Vascular, Nerve, Visceral and Musculoskeletal Tissue Engineering

**DOI:** 10.3390/polym12030620

**Published:** 2020-03-09

**Authors:** Anatolii Abalymov, Bogdan Parakhonskiy, Andre G. Skirtach

**Affiliations:** Department of Biotechnology, Faculty of Bioscience Engineering, Ghent University, 9000 Ghent, Belgium

**Keywords:** tissue engineering, interstitial, connective, vascular, nervous, visceral, musculoskeletal, polymers, hybrid, hydrogels, layer-by-layer, cells

## Abstract

In this review, materials based on polymers and hybrids possessing both organic and inorganic contents for repairing or facilitating cell growth in tissue engineering are discussed. Pure polymer based biomaterials are predominantly used to target soft tissues. Stipulated by possibilities of tuning the composition and concentration of their inorganic content, hybrid materials allow to mimic properties of various types of harder tissues. That leads to the concept of “one-matches-all” referring to materials possessing the same polymeric base, but different inorganic content to enable tissue growth and repair, proliferation of cells, and the formation of the ECM (extra cellular matrix). Furthermore, adding drug delivery carriers to coatings and scaffolds designed with such materials brings additional functionality by encapsulating active molecules, antibacterial agents, and growth factors. We discuss here materials and methods of their assembly from a general perspective together with their applications in various tissue engineering sub-areas: interstitial, connective, vascular, nervous, visceral and musculoskeletal tissues. The overall aims of this review are two-fold: (a) to describe the needs and opportunities in the field of bio-medicine, which should be useful for material scientists, and (b) to present capabilities and resources available in the area of materials, which should be of interest for biologists and medical doctors.

## 1. Introduction

Except for the gut epithelium [1], cornea [2], skin [3], and liver [4], regeneration of adult tissue after an injury or degeneration is is an inefficient process. This is addressed by designing and applying innovative biomaterials targeting regeneration of tissues. Although promising data have been obtained in laboratories, some hurdles limiting the clinical translation of implants still remain including cell survival, migration, and integration [5]. Polymer-based coatings perfectly fill the niche of soft-tissue repair, because of the similarity between their mechanical, physico-chemical, and physiological properties and those of tissue.

There are different methods of assembling polymers into coatings and scaffolds. These include, for example, hydrogels, layer-by-layer (LbL), Langmuir-Blodgett films, polymer brushes, etc. Hydrogels represent a broad research area with already existing practical applications; their biggest advantages include possibilities of incorporation of various particles, molecules, entities into the volume occupied by hydrogels. LbL assembly, where positively and negatively charged polyelectrolytes are sequentially applied, is often regarded as the substitute for Langmuir-Blodgett films, which are produced by drawing the substrate at a steady speed from a solution containing polymers. Flexibility of design of the coatings using LbL is one of their biggest advantages. Polymer brushes is another approach that allows to tailor functionalize the coatings, where their responsiveness to external stimuli is one of their biggest advantages.

Polymers stand out as a special class of materials due to the flexibility of design of their blocks and various functionalities, which is brought by their versatile assembly or by introducing additional side-groups or blocks, in the case of block-co-polymers. In this regard, both synthetic and biocompatible polymers are available, and polymer engineering represents a growing area tailoring the needs often driven by applications. With all advantages and the flexibility of the design, there is one significant weakness of polymers—the softness. In regard with tissue engineering, that translates to the fact that some materials can match predominantly soft tissue, but it is difficult to tune mechanical properties of polymers to match the hardness of bones, tendons, etc. And since mechanical properties of materials affect and even drive cell and tissue growth, enhancement of mechanical properties of polymers and biomaterials is essential. Chemical crosslinking can be applied to address these challenges, but that solves problems only partially.

To solve these challenges the so-called hybrid materials [6] are used, which possess both inorganic and organic contents. Increasingly, hybrid materials are used in designing the extracellular matrix. The hybrid matrix design for various tissue engineering applications should meet bio-medical requirements. On the one hand, it needs to be biocompatible and biodegradable, which is achieved through the polymeric or organic content, while on the other hand, stability and possess mechanical properties are needed to match those of a desired tissue—this is achieved through tuning the type, concentration, and distribution of the inorganic content. Each organ has different types of tissues that require different functionalization of polymer matrices [7]. For example, in the case of a tendon, one needs to create a hybrid material that should have excellent mechanical properties and allow integration of both bone and muscle tissue [6]. For such an organ as the lungs, there are different softness areas within the range to be matched by polymers, so tuning mechanical properties of materials is less essential. But in the case of the trachea, the mechanical properties play a very important role, and for the alveoli—the transport of substances and the rate of implant angiogenesis [8].

For hard tissue repair hybrid biomaterials are increasingly used [9]. In general, polymer-based hybrid biomaterials: (1) have extensive processing flexibility, tunable mechanical properties, and biodegradability; and (2) possess individual properties and controlled functions, which can be intelligently tailored to actively fill-in the needs of the regeneration in tissue engineering [10,11]. Another development in the area of polymer-based biomaterials for regeneration of hard and soft tissue are smart biomaterials [12]. Such smart functionality can be invoked by chemical, physical and biological stimuli [13,14,15]. Various stimuli allow controlling appropriate biodegradation profiles: as it was shown, for example, for polymers suspended in a solution [16], chemical triggering for bringing polymers together [17] or laser remote release of materials immobilized on bio-coating [18,19,20] through a localized temperature rise [21,22,23].

The addition of particles with encapsulated biomolecules allowed the implementation of the concept of both hardness modification and drug delivery with the coatings [24,25]. Such modification of the coatings can be well suited for applications in pharmacology [26], where delivery of molecules is just as essential as control over physico-chemical properties of films [27,28] It is interesting to note that functionalization of hybrid assemblies [29] can be made with micro- or nano- particles, where the former size is more suited for thicker polymeric films and hydrogels [30,31], while smaller nanoparticles are better suited for tuning mechanical properties of less gelly films [32]. It is envisioned that the distribution [33] of such particles will allow to fully control the softness-hardness of films and other physico-chemical properties.

In addition to inorganic particles, various capsules provide the means of yet another type of modification of polymer-based films to promote the interaction with cells. There, drug or biomolecule delivery can be implemented in addition to mechanical properties control. Incorporation of capsules was applied earlier [34] using hyaluronic acid/poly-L-lysine exponentially grown films [35]. Such a versatile example allowed not only incorporation of capsules into films, but also remote activation of capsules by laser irradiation [36] and thus release from the coatings [37]. Polyelectrolyte multilayer capsules possessing growth factor and incorporated into polymeric films were shown to promote stem cell growth to bone cells [38]. Furthermore, various types of polyelectrolyte multilayer films can be functionalized with capsules and particles often in the form of calcium carbonate, as shown by Li and co-workers [39]. Microencapsulation has been shown not only to bring additional functionalities, but also to be used for controlling the rate of release [16]. Furthermore, addition of microcapsules allows brining-in sensor functionalities [40,41]. Mechanical properties of microcapsules can be tuned by studying deformation of capsules [42] together with release followed by fluorescence microscopy [43] so that the shape microcapsules can be changed [44] corresponding to the induced mechanical deformations.

Here, we discuss state of art and recent developments in application of innovative polymer-based and hybrid bio-materials addressing the needs of different tissue engineering areas. On the one hand, an overview of pure polymer-based as well as hybrid materials is made together with the methods of their assembly. While on the other hand, applications of these materials for interstitial and connective (tendon and the skin), vascular, nervous, visceral (lungs, kidney, liver), and musculoskeletal (cardiac, skeletal, cartilage and bones) tissues are discussed. A particular attention is paid to mechanical properties of coatings, since they most often enable application of these materials for different areas of tissue engineering. A summary of main properties of above mentioned materials is presented together with their applications in respective tissue engineering areas.

## 2. Polymers in Tissue Engineering

Nowadays, tissue engineering is one of the most interdisciplinary scientific research areas. Indeed, chemistry, materials science, engineering, physics as well as biology and medicine are combined in this area. Some key directions in tissue engineering is development of suitable bio-interfaces, their integration in appropriate tissues and control over their properties, for example, for release of active bio-molecules. This is achieved through an appropriate combination of a matrix (e.g., polymers) and bioactive molecules (e.g., growth factors or enzymes) and cells applied to specific tissues and organs.

The polymer matrix is one of the most important components of hybrid biomaterial. The matrix may consist of synthetic (polyglycolic acid (PGA), polylactic acid (PLA), polycaprolactone (PCL), poly (N-isopropylacrylamide) (PNIPAM)) [24,45,46,47] or natural (chitosan, alginate, collagen, hyaluronic acid, gellan gum, gelatin) polymers [48,49,50,51].

Suitable polymer scaffolds for tissue engineering should be biodegradable, biocompatible, and they should not cause mutations and a strong immune response [52]. In addition, scaffolds need to mechanically correspond to cells and tissues (to accompany differentiation of stem cells into the required cell type and stimulate proliferation) [53]. Another essential functionality of scaffolds is biodegradation, which can take place by means of enzymes. In the case of natural polymers, they are metabolized and removed from the body after some time [54].

### 2.1. Syntetic Polymers

Polyglycolic Acid (PGA) is one of the simplest linear, aliphatic polymers. Biodegradability and biocompatibility make this polymer convenient for creating biomaterials. PGA can be made by opening the glycolic acid ring [55]. PGA has been used more than once to create fiber matrices for tissue engineering [56]. PGA shows a very fast vascularization result within a week after implantation [57]. Hybrid materials are also successfully created on the basis of this polymer, for example PGA/hyaluronic acid to restore cartilage, PGA/collagen to stimulate vascularization [58].

Polylactic acid (PLA) is also an aliphatic polymer, which is used to create biomaterials owing to its biocompatibility. PLA is made from lactide ring opening polymerization [59]. It is often used in the form of parts for fastening bone tissues (screws and pins) [60]. In regard with hybrid implants, it is worth noting that the chitosan/PLA nanofibers coated with hydroxyapatite were used to accelerate bone tissue regeneration [61]. Also, PLA in various proportions with thermoplastic polyurethane (TPU) has exhibited good results in tissue engineering applications [62].

One of the most common synthetic polymers is polycaprolactone (PCL). PCL is a biocompatible and biodegradable polyester. The synthesis of PCL is carried out using an open-ring polymerization [63]. The main type of matrix for this polymer is nano and microfiber scaffold, which is formed by the electrospinning method [64]. A huge amount of hybrid biomaterials is formed from PCL for various types of tissues and organs. The only problem with PCL is a relatively poor cell adhesion [65]. Therefore, there is a large number of functionalization of this material from collagen to calcium carbonate [66,67]. Hybrid PCL materials can be applied to almost all types of tissues and organs from tendons to lungs [68,69].

PNIPAM is a thermo-responsive polymer, which undergoes a phase transition from hydrated to dehydrated state at 32 °C. Applications in tissue engineering have been developed, where a physical stimulus, temperature, has been used for changing the properties of coatings containing this. Specifically, a controllable cell attachment/detachment has been performed by increasing temperature of a solution containing cells, while the state of films, their hydration state and their mechanical properties have been characterized by atomic force microscopy (AFM) [70].

A relatively large number of charged polyelectrolytes exist for LbL applications. Although one of the most studied pair in the area of LbL is polyallylamine hydrochloride (PAH) and poly-styrene sulfonate (PSS), other polyelectrolytes have been also extensively used. These include: polydimethydiallylammonium chloride (PDADMAC), polyvinyl alcohol (PVA), poly-acrylic acid (PAA), polyethylene imine (PEI). In addition to these synthetic polymers, some natural/biodegradable polymers have been used including dextran sulfate, polylactic acid (PLA), poly-arginine, poly-L-lysine, hyaluronic acid (HA). Other natural polymers are discussed in the following section.

### 2.2. Natural Polymers

Alginate is a polysaccharide derived from brown algae. Due to specific structure, alginate is applicable in medicine. An alginate gel is formed by binding to two (or higher) valence ions [71]. An advantage of alginate is the capability to easily encapsulate animal cells [72]. But pure alginate has very poor adhesion properties, and only a modified alginate gel can adsorb cells on itself [71]. Composites of alginate with hydroxyapatite and bioglass are often used for bone tissue engineering [73]. Alginate is also often used to treat skin, including burn treatment [74,75].

Collagen is the main component of connective tissue and the most abundant protein in mammals. Collagen makes up 25% to 45% of the proteins in the whole body. The collagen molecule is a left-handed helix of three α-chains. This formation is known as tropocollagen [76]. Since collagen is a product of an animal body, it is metabolized quickly without causing an inflammatory reaction [66]. Typically, collagen is used to create hybrid biomaterials to improve cells adhesion [77].

Chitosan is a derivative of chitin, which originates from the shell of shrimps, molluscs, fungi. Boiling chitin in potassium hydroxide results in chitosan. Chitin is also biodegradable and does not cause a strong response from the immune system [78]. It has huge applications from medicine and biomaterials [61]. There are some limitations when working with chitosan, since the dissolution of chitosan occurs only with increased acidity, which makes some restrictions when working with cells and pH-dependent molecules [79].

Hyaluronic acid (HA) is a naturally occurring, anionic glycosaminoglycan widely used in up-keep products to preserve moisture and smoothness of skin. It is also found in extracellular matrix (ECM) making it often sought for material in tissue engineering [80,81]. Coatings with HA are rather soft, which prompted application of cross-linking to it [82]. For example, HA is one of the most important constituents of the so-called exponentially grown LbL films [83].

Cellulose belongs to the family of linear chain polysaccharides. Peculiarly, cellulose has been also studied for applications in tissue engineering [84]. Biodegradability of cellulose can be an issue for application in tissue engineering, because it either does not degrade or degrades very slow (over 60 weeks) [85].

### 2.3. Assembly of Polymers into Coatings and Films: Hydrogels, LbL, Langmuir-Blodgett Films, Polymer Brushes, 3D Printing

Hydrogels are very useful but, at the same time, some of the softest bio-materials [86,87]. Functionalization of hydrogels with inorganic particles leads to enhancement of their mechanical properties through, for example, additional bio-mineralization [50] and incorporation of possibilities to release encapsulated materials or antibacterial properties [88].

Discussed by Decher, LbL films have seen two decades of rapid development [89,90,91,92,93,94], where both versatile planar layers and fully suspended polymeric capsules have been established. Typically, LbL films are quite thin, but a special design of the so-called exponentially grown films leads to micrometer thick gel-like films [95], which are applicable for tissue engineering. In this regard, the deposition time affects the uptake of polymers [96] and water [97]. The flexibility of the design of LbL coatings enables extensive opportunities including those in tissue engineering, where vascular patches [98] have been designed. Detachable cell sheets for tissue engineering have been reported [99].

Another desirable feature for tissue engineering is patterning, which was also reported for LbL structures [100]. Differentiation of stem cells has been obtained with LbL coatings [101]. Adhesion of cells can be promoted either by patterning [102], chemically patterning the LbL sources [103] or by adjusting mechanical properties [104]. Adjusting the mechanical properties, which can be done by addition of gold nanoparticles on the surface, allows not only for controlling cell adhesion, but also to controllably embed and pattern particle [105,106].

Composite LbL films containing inorganic nanoparticles allow one to control cells adhesion, while also giving the material the required properties (stiffness, antibacterial properties, etc.) [107]. Polymer brushes represent polymers attached or tethered to a surface [108]. To distinguish between, for example planar LbL layers, the density of polymers constituting polymer brushes is said to the higher than the gyration radius of polymers. Application in tissue engineering can be also thought with polymer brushes [109]. Control of rigidity, reversible physico-chemical properties and stimuli-responsiveness [109] are noted as some distinct properties of polymer brushes making them unique materials in tissue engineering.

In addition, 3D printing is a rapidly expanding area due to availability of new instruments and materials [110,111,112]. Several approaches to 3D printing are available including fused deposition modelling, selective laser sintering, stereolithography, near-field electrospinning, and bioprinting, where inkjet printing, laser-assisted 3D printing, extrusion [113,114].

### 2.4. Pure Polymeric Coatings Targeting Different Tissues

Numerous applications in tissue engineering are developed using the so-called hybrid materials, where polymers are supplemented by inorganic particles. But this is not always the case, and there are still applications where pure polymers are used for constructing scaffolds or other structures for tissue engineering.

One of the most important criteria for selecting pure polymers for various applications in different tissues and organs is their mechanical properties [115,116,117]. Indeed, human tissues have their own rigidity depending on the type of cells and their structural organization [118]. Peculiarly, mechanical properties of tissues range from kPa to GPa, based on data for arterial walls [119], brain [120], breast [121], bone [121], cartilage [122], cornea [123], heart [124], kidney [125], liver [125], prostate [126], vein [127], skin [127] and tendon [128]). A summary of these data is given in Figure 1. Analyzing these data, it can be seen that chest and liver have a very low modulus of elasticity of about 1 kPa, which is similar to that of polyacrylamide. Bones, on the other hand, have relatively speaking extremely high rigidity (of the order of 10 GPa), which can be matched by poly (methylactrylate) with rigidity in the range of 2 to 4 GPa [129].

Furthermore, such natural polymers as collagen, gelatin, alginate and agarose gels have good biocompatibility, but they are only suitable for soft tissues [130]. Synthetic polymers have a fairly wide range of stiffnesses, but they can cause inflammation in the body, and often have problems with cell adhesion [131].

**Figure 1 polymers-12-00620-f001:**
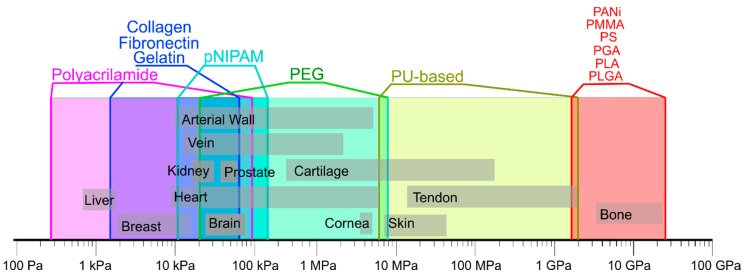
Mechanical properties of natural tissues and polymers. Data are composed based on data from the following publications [25,131,132].

It is these aspects that limit the use of pure polymers for tissue engineering prompting to use of hybrid materials, which are overall better suited for fine-tuning of relevant properties of implants [131], and can be also used for drug delivery through incorporated particles. In the following section, hybrid materials are discussed in regard with their applications in specific tissue engineering areas [25].

## 3. Hybrid Biomaterials for Interstitial and Connective Tissues Reconstruction

### 3.1. Hybrid Biomaterials Applied for Tendon

The transfer of power between the musculoskeletal tissues is due to tendons and ligaments. The structure of the collagen fibrils that make up the tendons ensures their strength and elasticity. The main components of tendons are fibrillar, fibril-associated (I, III, IV, V), beaded filament-forming (VI), and network-forming collagen (IV), a large amount of elastin and decorin proteins are also present, expression of chondroitin and proteoglycans by cells provides for swelling in the aquatic environment. The orientation of cells inside the tendons occurs around collagen fibers—this positively affects also the mechanical properties [128,133]. The insertion of the tendon in the bone consists of three continuous regions, in which the soft tissue transforms in fibrocartilage and, then, in a bone. The fibro-cartilaginous region is further divided into calcified and noncalcified regions. These complex transitions are essential for the biological fixation of the tendon and exhibit a gradient of structural and mechanical properties (>310 kPa) to transmit complex loads between the tendon and the bone insertion site [134]. To mimic these structures, hybrid materials with engineered matrix inhomogeneities and structures are required. A wide variety of such synthetic and natural polymers as PCL, PGA, polyurethane, collagen, and silk fibroin, and a variety of scaffold production methods have been used to develop functional matrices for tendon and ligament tissue [66,135,136]. The base for tendon implants is made using polymers matrix, which can be obtained by electrospinning due to their architecture and good mechanical properties, but in connection with poor cell adhesion of the pure polymers and tendon construction often use collagen coating. Also, fibro-cartilaginous region formation stimulated by calcium-based nano and microparticles of hydroxyapatite [68,137]. A striking example of a solution to this problem is the work of Savelyeva et al. They coated PCL nanofibers by CaCO_3_ particles loaded tannic acid. It was reported that particles improve cell adhesion on the surface of the PCL implant, and when connected to the bone, after a while passed into the form of hydroxyapatite, which significantly improved the integration of the implant on both sides [138] (Figure 2).

### 3.2. Hybrid Biomaterials Applied for Skin

Knowledge of native tissue is one of the major factors in successful development of biomaterials for regenerative medicine [139]. Skin is the biggest organ of the human body, which has a lot of functions, some of them are protection, sensation, thermoregulation, control of evaporation etc. Skin includes two anatomically, functionally, and developmentally distinct tissues: the epidermis, which has a role of barrier to infection and regulator of transdermal water loss, and the dermis, which has mechano and thermoreceptors, hair follicles, different glands and vessels [140]. When developing biomaterials for skin regeneration, it is important to remember that the main components of a successful biomaterial are ECM, cells and bioactive molecules that help cell growth or differentiation [141,142].

In the case of the development of skin implants, scientists can start using not only pure polymers, because the restoration of the skin occurs much faster if one uses various combinations of polymers with nano- [143] or micro- [67] particles and bioactive substances [144]. Hydrogels can be also used, because, although they are soft, skin represents also a soft tissue [145,146,147,148]. It should be added that the Young’s modulus of skin is in the range from 4.5 till 8 kPa [134], that is why materials should match these values. Mechanically affected regions of skin are usually located near to the surface that may lead to a rejection of implants due to contamination and inflammation [149]. One solution to this potential problem is a modification of polymer hydrogels and fibers by microcontainers or microcapsules with, for example, tannic acid, which has antimicrobial properties. This would resist fibrosis and would improve angiogenesis of tissues [138,148,150,151]. It should be noted that signals generated by VEGF and TGF play an important role for proper regeneration of skin tissue and healing of wounds [144,147,152,153].

## 4. Hybrid Biomaterials for Vascular Tissue Reconstruction

Tissue engineering of vascular tissues is one of the major application areas of tissue engineering, because cardiovascular disease is linked to one of the highest number of death cases. Treatment for cardiovascular diseases can include a complete replacement of a destroyed vessel with an artificial one. Delivery of oxygen-rich blood from the heart to the different tissues is particularly important for blood vessels [154].

Arteries represent a muscular tube lined by smooth tissue. It has three layers: the intima, which presents a single layer of flat cells, the media, which present the thickest layer in arteries and rich in vascular smooth muscle, and the adventitia, which represent the thickest layer in veins and entirely made of connective tissue [155]. Such a vessel architecture allows you to experience high pressure from the heart (from 1600 till 8250 mmHg) [156].

One of the useful polymers for vessels engineering of medium-large diameter blood vessels is expanded polytetrafluoroethylene (ePTFE), because of its high crystallinity and hydrophobicity. These two characteristics of ePTFE ensure the prevention of hydrolysis. But for small diameter vessels using ePTFE can be dangerous because of intimal hyperplasia, calcification, and thrombosis [157]. Therefore, a biodegradable implant is considered the best option, which will be replaced by vascular cells, preserving the architecture of the vessel, but without losing its mechanical properties [158].

In most cases, coatings of implants for vessels should contain bioactive molecules, which speed up vessel’s integration. One of them is vascular endothelial growth factor (VEGF)-a family of signal proteins produced by cells for stimulating the formation of blood vessels. Interesting work in vessels implant area is done by the Gonzalez-garcía group. In that work, poly (ethyl acrylate) was shown to lead to the formation of fibronectin networks, where VEGF-presenting areas were generated, (Figure 3), which promote vascular cell response. Such a system allows to use extremely low doses of the growth factors, while at the same time improves vascularization. As such, it is robust and efficient approach to improve vascularization of the tissue and in can be implemented in 2D and 3D scaffolds [159].

## 5. Hybrid Biomaterials for Nervous Tissues Reconstruction

The causes of death of neurons, astrocytes, and oligodendrocytes may be apoptotic or necrotic. The causes of death and poor functionalization can also be axon damage, demyelination, ischemia, oxidative damage, and inflammation. The main problem of neural tissue regeneration is the complete lack of stimulating growth and migration factors, but there are factors responsible for inhibiting axon growth. In this regard, the damaged area quickly forms a healed scar, which is a chemical and physical barrier to regeneration [160]. Therapy of damaged areas is very difficult due to the architecture of the central nervous system. Cell transplantation is also difficult to achieve due to adverse conditions at the lesion sites. These factors make neural tissue engineering one of the most complex. Nowadays, the area of nanofibers consisting of natural and synthetic polymers receives substantial scientific interest.

One of the main approaches of tissue engineering of nerves is the creation of nerve conduits and the replacement of damaged ones. New conduits can be synthetic, biological or hybrid, which are biosynthetic. Synthetic polymers such as polyester, polyurethane, and polyols are widely used to create conduits. Also, biological polymers can be used to create them: polyesters, proteins, and polysaccharides [161]. Nerve conduits can be of three types: tubular, fibrous and matrix. A striking example is the creation of a hybrid fibrous nerve conduit, which consists of PCL and gelatin. Fibrous PCL has an excellent architecture for cell migration, and gelatin stabilizes the electrically pumped PCL fibers and provides good cell adhesion. The growth of neurons in the hybrid conduit showed a better differentiation and migration of neurons than that for a channel consisting of only one component [162].

Also, for good survival and treatment of damaged nerve sections, growth factors such as nerve growth factor (NGF) [163], neurotrophin-3 (NT-3) [164,165] and brain-derived neurotrophic factor (BDNF) [166] should be included in biomaterials.

Hydrogel implants consisting of agarose, hyaluronic and polyethylene glycol and containing a ciliary neurotrophic factor (CNTF), sonic hedgehog (SHH) [167], vascular endothelial growth factor-165 (VEGF165) [168], and EGF [169], as well as cell-adhesive arginylglycylaspartic acid (RGD peptides) [169] have been created. Martino et al. showed the incorporation of VEGF, PDGF, and bone morphogenetic protein-2 growth factor into the same binding site as integrin on fibrin fibers. This approach increased the biological activity of the growth factor [170].

Mechanical properties are some of the most important characteristics for nerve regeneration. A good flexibility of the implant helps to avoid axon destruction at the bending of the nerve [171,172]. An implant that is too rigid can compress the growing tissue [173]. But a substantially weak mechanical strength of the matrix can lead to breakage, rupture, and kinks of the nerve. Thus, an optimal polymer or copolymer ratio must be ascertained through experimental studies [172]. Using hydrogels as one of the components for creating hybrid implants, it is worth emphasizing the importance of swelling. As a result of swelling, axons are compressed, and their growth is disturbed. Swelling occurs due to the accumulation of conduit decomposition products, and it is controlled by changing the ratio of the polymers [174]. Optimal degradation should be uniform along the entire length of the implant, because, too fast degradation causes inflammation and swelling, and too slow degradation causes compression [172].

## 6. Hybrid Biomaterials for Visceral Tissue Reconstruction

### 6.1. Hybrid Biomaterials Applied for Lungs

To create hybrid materials for tissue engineering of the lungs, one must have a good understanding of the anatomy and physiology of this organ [175]. Anatomically, the lungs are divided into two sections—the proximal and distal airways. The proximal respiratory (>2 mm internal diameter) consists of the pseudostratified epithelium. The pseudostratified epithelium is multilayered: the basal layer (discontinuous layer of the basal cells) and the luminal layer (ciliated cells, Clara-like cells (secretory club cells), human goblet cells, and neuroendocrine cells that aggregate to form neuroendocrine bodies). The distal airway (small bronchi and bronchioles) consists of a single layered epithelium containing Clara cells, ciliated cells, neuroendocrine cells, and p631Krt51 basal cells [176].

Alveoli are located in the most distal part of the lung and consist of two types of epithelial cells: alveolar type I cells (AEC I) and cuboidal alveolar type II cells (AEC II). Alveolar type I cell provide a gas exchange surface. Cuboidal alveolar cells containing secretory vesicles filled with surfactant [176]. Park et al. have hypothesized that a biodegradable implant that “disappears” over time will have a good effect on the regeneration of tracheal tissue and facilitate the approximation of native tissue (Figure 4A) [8]. The authors used a hybrid implant consisting of L-lactide and ε-caprolactone, which was synthesized by ring-opening polymerization (P [LA/CL]). Such a hybrid matrix decomposes in 2–3 months in vivo by hydrolysis. A special advantage is the high porosity of such a matrix—80%. Pores have a size of approximately 20 to 100 μm, which allows cells to migrate into the implant. It was also possible to strengthen the matrix due to the gelatin coating, which also improved cell adhesion on the surface [69,177]. This is not the only example of the successful use of hybrid matrices. Also, the gelatin matrix coated with PLCL/TGF-B1 with cultured chondrocytes on the surface showed good results in vivo experiments (Figure 4B). Within 8 weeks after implantation, the matrix retained its mechanical properties, despite the gradual replacement by native tracheal tissues (Figure 4B) [8].

Higuita-Castro et al. have been successful in developing a platform that mimics the micro/nanoscale architecture of the alveolar environment. They used this system to study the effect of the structural properties of the ECM on damaged epithelial cells during the reopening of the airway (Figure 3C). Gelatin coated PCL nanofibres are used to adjust Young’s modulus. As a result, it was found that the stiffness of hybrid matrices significantly affects the barrier function of the epithelium/endothelium. An increase in the rigidity or density of the matrix led to a change in the structure of the cytoskeleton and a change in cell permeability [178].

### 6.2. Hybrid Biomaterials Applied for Kidneys

Kidneys perform the following essential functions: excretion and filtration of metabolites, regulation of the electrolytes, fluids, the balance of acids and bases, and maintenance of the production of red blood cells (RBC). The renin-angiotensin-aldosterone system of the kidneys is also responsible for regulating blood pressure [179].

An important factor in tissue regeneration in the kidneys is the cellular microenvironment, which can be adjusted by the proper engineering of hybrid materials. One example of success using hybrid material is hyaluronic acid/collagen crosslinked with PEGDA and loaded stromal cell-derived factor-1 (SDF-1). This hybrid material was also used as an endothelial progenitor for cell delivery system. There, hydrogel protects EPCs from cytotoxic insult. SDF-1 loaded into hydrogel stimulated proliferation and increased cell engraftment in kidneys [180,181]. It should be noted that such strategies for using hybrid polymer hydrogels as cell carriers for kidney reparation are used quite often [182,183,184,185].

Lih et al. developed a PLGA matrix coated with Mg(OH)_2_ and acellular ECM. Mg(OH)_2_ can affect the acidity of the cellular microenvironment, which changes during the decomposition of the PLGA. As a result of reducing the acidity of the microenvironment, the inflammatory effect is also reduced. The ECM also improves the normal biological function of cells on the matrix [186].

Growth factors are a promising component of polymer-based hybrid materials. Hybrid materials can also be used for kidney tissue, as it is closely related to kidney regeneration in renal failure [186]. The acellular ECM could promote the normal biological function of kidney cells.

Classical methods for coating polymers with growth factors are adsorption and chemical bonding [159,187]. The healing process needs the replacement of damaged tubular cells to restore the connection of the renal epithelium. This process includes some growth factors that are provided in renal tissues. The growth factors include epidermal growth factor (EGF), a potent proximal tubule cell mitogen [188]; transforming growth factor-α (TGF-α), a participant in the reconstruction of the injured proximal tubule via the EGF receptor [188]; insulin-like growth factor 1 (IGF-1); fibroblast growth factor (FGF), and hepatocyte growth factor (HGF) [189].

### 6.3. Hybrid Biomaterials Applied for Liver

The largest internal organ in a body is the liver, which performs hundreds of different functions that help the normal body homeostasis [190]. Damage to tissues or cells that entail hepatic pathology provokes a change in physiology and metabolic activity. Although the liver has a high regenerative ability and can completely restore its mass and function, some disorders affect the physiology so much that therapy is required to restore its functions. Typically, these effects cause hepatitis, cancer and various toxins, such as drugs and alcohol [191].

Hybrid polymeric materials are actively used for tissue engineering of the liver too. Due to them, there is the possibility of current adjustment of cell adhesion and migration. Earlier, natural (cellulose, chitosan, gelatin, alginate, collagen, fibrin, heparin, hyaluronic acid) and synthetic (PVA, PLA, poly- (N-isopropylacrylamide), poly (lactide-co-glycolic acid) and PCL) polymers have been applied for tissue engineering of the liver. Besides, growth factors and specific enzymes have been be used as one of the components of the hybrid matrix [192,193,194].

Currently, hybrid polymeric materials are already actively used in tissue engineering of the liver [195]. The adhesion peptides RGD or YIGSR significantly improved the adhesion of hepatocytes on the surface of PCL and PLA, compared with pure polymers [196]. Lee et al. showed a polymer poly (ethylene glycol) diacrylate (PEGDA)/hyaluronic acid-coated with a synthetic peptide Gly-Arg-Gly-Asp-Ser (GRGDS). This semi-permeable gel supported the differentiation of hepatocytes for 16 days [197]. The 3D sandwich culture of hepatocytes between the top and bottom layers maintained their functional state over a period of 14 days. Such cultures were even faster than collagen sandwich cultures in the rate of tissue formation [198]. Qui et al. modified PCL scaffold with galactosylated chitosan and showed an improved hepatic functionality over a period of 7 days as compared to a 2D monolayer or PCL 3D scaffold [199].

## 7. Hybrid Biomaterials for Musculoskeletal Tissue Reconstruction

The replacement of bones, cartilage and skeletal muscle are the most essential goals of musculoskeletal tissue engineering. Although the cell type and even their source as well as materials themselves influence the tissue growth, new developments, recent progress in this area allows for effective cell cultivation and growth. The following polymers with their particular advantages have been used in musculoskeletal tissue engineering: collagen (cell differentiation), Matrigel (proliferation), Sylgard (matrix formation), fibrin gels (cell survival), PGA (vascularization), alginate (adhesion), PCL (adhesion), hyaluronic acid (immunogenicity) [200]. Just like in other application areas, patterning of gels [201,202] is essential as well as application of natural and synthetic polymers [203] is essential for musculoskeletal tissue engineering. Overall, the designed scaffolds should withstand forces of external environment and match or mimic the properties of cells surrounding them.

### 7.1. Hybrid Biomaterials Applied for Cardiac Muscle

One of the leading causes of death in developed countries is cardiovascular disease. The regenerative capabilities of the heart muscle are extremely limited; as a result, millions of cardiomyocytes die after myocardial infarction [204,205]. After a myocardial infarction, a scar forms within 4–6 weeks. This collagen scar, consisting of fibroblasts and endothelial cells, significantly reduces the contractile function of the heart. Such a change leads to heart failure. An appropriate combination of polymers, bioactive molecules and cells can help restore cardiac muscle function after such injuries [206].

A striking example of an application of a hybrid material is the use of an elastomeric framework of nanofibers treated with macromer-thick layers of acrylate poly (glycerolebacinate) (Acr-PGS) obtained by electrospinning. Gelatin was used as the polymer, which stimulated the adhesiveness of cells onto the scaffolds. The obtained scaffolds have different mechanical properties, depending on the ratio of the polymers used (tensile modulus from 60 kPa to 1 MPa) and degradation rate (weight loss from about 40 to 70% by the beginning of the 4th month). At different polymer ratios, a difference was also observed in the chemical response of the recipient to implantation [207]. A heart patch was constructed from poly (glycerol subacute) and was found to be sufficiently active even without gelatin for cell adhesion for more than three months [208].

### 7.2. Hybrid Biomaterials Applied for Skeletal Muscles

Skeletal muscles recover well in response to minor injuries, but serious damage can cause fatal consequences in muscle activity if therapy is not done on time. Hybrid biomaterials that provide chemical and physical signals can help improve muscle regeneration. The two main factors for restoring muscle work are the angionization and innervation of the implanted scaffold. An important factor in the survival of an implant after transplantation into muscle tissue is its vascularization. Often, to accelerate this process, implant pre-vascularization is carried out, for example, a matrix consisting of fibrin/poly (lactic-co-glycolic acid) is pre-seeded with a mix of fibroblasts and myoblasts. The matrix with cells is transplanted to the recipient. After a couple of weeks, angiogenesis and anastomosis of the vascular system of the recipient with implant begin. After that, this matrix is transferred to the damaged area, preserving the vasculature [209]. Also, polymers are coated with arginine and laminin before implantation, which improves the clustering of acetylcholine receptors [210,211]. This procedure improves subsequent tissue innervation [211]. When cultured with neurons, such myofibrils form acetylcholine receptors at the border of two cell types. When exposed to glutamic acid, one could observe shortening [212,213].

### 7.3. Hybrid Biomaterials Applied for Cartilage

Cartilage defects are some of the most common problems of orthopedics, while similar pathologies result from trauma, aging, etc. Cartilage has a very low ability to regenerate since it only consists of chondrocytes. Cartilage tissue lacks innervation and vascularization [214]. One of the most used form of polymers for the regeneration of cartilage tissue is a hydrogel. Hydrogel implantation procedures can be minimally invasive (injection or arthroscopy). The advantage of using polymer hydrogels is its properties: swelling and lubrication [215,216,217]. Mechanics can be also added to this list provided that the hardness can be tuned with an inorganic content in hybrid hydrogels. These properties are also inherent in native cartilage tissue, which allows the implanted material to successfully mimic the tissue. The morphology of the cells inside the hydrogel is mainly spherical, which is characteristic of chondrocytes [218]. One of the hybrid hydrogels used to regenerate cartilage is polyethylene glycol (PEG)-based hydrogel with methacrylate groups for photo-crosslinking [215]. When growth factors are added to such hydrogels, a significant acceleration of the cartilage formation of tissue by stem cells is observed [219].

### 7.4. Hybrid Biomaterials Applied for Bones

One type of tissue with an inherent regenerative capability is bone tissue [220]. However, if the disturbances or injuries are too extensive or the regeneration process has been interrupted by a new exposure, major healing problems may arise. There is also a big problem with bone regeneration for old people due to osteoporosis. In all these cases, an exterior intervention is required [221]. The bone mainly consists of connective tissue, which is continuously undergoing the destruction of old sites and the creation of new ones, by osteoclasts and osteoblasts, respectively. This occurs due to high physical stress on these tissues [222]. For material scientists, bone is a matrix composed of polymers (collagen) and ceramics (hydroxyapatite). Therefore, the use of hydrogel and ceramics can improve implant engraftment both at the cellular (good adhesion and fast differentiation) and at the organism level (mechanical properties required for bone tissue) [223].

An important parameter for creating hybrid materials for bone regeneration is the ability to promote primary biomineralization. For example, when using PCL fibrous matrices (fiber Ø 50 μm) coated with calcium phosphate, biomineralization occurs much faster than that on pure PCL. Such polymer coatings increase alkaline phosphatase activity and accelerate in vivo bone formation [224]. It was also demonstrated that implantation of only ceramics did not lead to an acceleration of osteogenesis [225]. This indicates the need for a stable surface. That is why hybrid materials are so in demand for bone therapy [226,227].

Grafting of enzymes can be also used for promoting the cell adhesion. In this regard, it was recently shown by Muderrisogly et al. that an enzyme (alkaline phosphatase, ALP) can be immobilized onto a titanium surface with a alginate hydrogel (Figure 5) [228]. This was carried out in several steps: (1) the mineralization of a titanium substrate by CaCO_3_ particles; (2) the adsorption of ALP enzymes; (3) the coating the substrate with the alginate hydrogel. Further, this structure is mineralized again, and ALP molecules are adsorbed on it. This design provides the implant with all the substances necessary for osteogenesis and improves the anti-corrosion properties of the matrix [229].

One can see from data discussed above that the invent of hybrid materials enable the so-called “all-in-one” concept. There, a polymer can be taken, modified by inorganic content, and tuned to match required properties of desired tissues or organs.

From discussion above, it is clear that pure polymer and hybrid materials have been used in various applications in general area of tissue engineering. These applications are often so distinct that they often are considered almost as stand-alone areas. Some selected applications are shown in Table 1 presenting tissue/organ type, coatings applied there, the type of polymers as well as targeted functionality.

## 8. Conclusions and Outlook

In summary, tissue engineering is an important interdisciplinary and multidisciplinary research area with an explosive growth potential. Two most important building blocks here are polymer-based and hybrid biomaterials. Innovations in the design of coatings, progress in their imaging and characterization as well as availability of new materials open opportunities for custom-tuned biomaterials suited for various tissue engineering sub-areas. Some of these materials are purely polymeric, yet they have been also shown to be directly applicable in tissues engineering. These polymeric materials are complementary to the so-called hybrids incorporating inorganic constituents together with organic (polymeric) matrix. These hybrids are increasingly used for matching the properties of various tissues and enabling thus the so-called “one-matches-all” concept. This refers to one or similar materials, which can be applied in various and most importantly different areas of tissue engineering. Besides inorganic particles or capsules in such hybrid coatings enable release of relevant bio-molecules. It can be alluded that although such tissues as tendon, skin, vascular and nerve tissue, lung, kidney, liver, cardiac and skeletal muscles, cartilage, bones) are very different, they can be all targeted with the same material base. An additional important functionality of hybrids is controlled release of encapsulated biomolecules.

Future directions of research are expected to experience further merger of biomaterials with bio-medical sciences, wherein scientists working in the former areas will be looking in more details to the needs in the bio-medicine. By the same token, doctors and biologists would benefit from new solutions and novel formulations offered by material scientists. Information on the properties of different tissues and state-of-art in their respective areas, overviewed here, will be helpful to design new materials and knowing already available ones. Further integration of biomaterials with tissue and organs will likely experience further interest to understanding their mechanical properties, surface functionalization, possibility of 3D and 4D printing technologies, physiological behavior. Additional in-vivo tests would play an increasing role in further integrating these two types of materials (organics and inorganics) and these two areas.

## Figures and Tables

**Figure 2 polymers-12-00620-f002:**
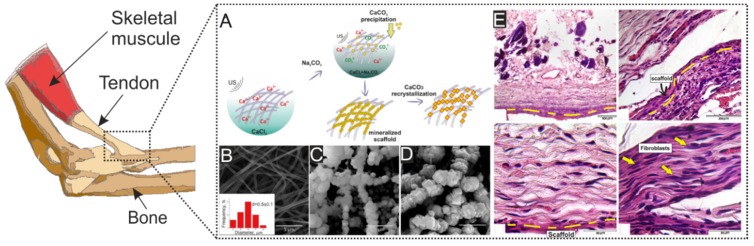
(**A**) Scheme of ultrasonic mineralization of polycaprolactone fibers and recrystallization of vaterite to calcite. (**B**) Microphotographs show pristine PCL fibers. (**C**) the mineralized scaffold based on PCL material. (**D**) process of recrystallization vaterite to calcite. (**E**) Micrograph of histological sections, 21 days after implantation in the withers of the mouse. Yellow dashed lines indicate the location of the matrix. Reprinted with permission from Elsevier, 2018 [138].

**Figure 3 polymers-12-00620-f003:**
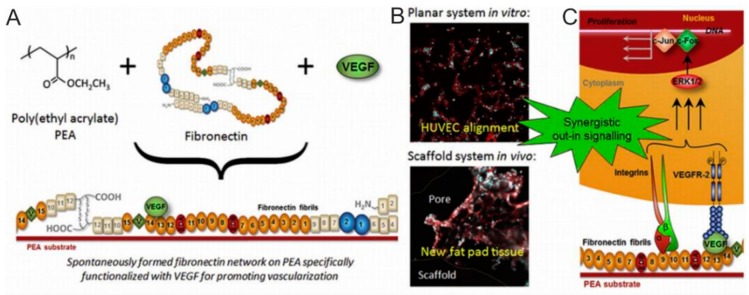
(**A**) Synergistic triggering of the integrin-VEGF signal, which is connected to the Poly (ethylene adipate) network. (**B**) In vitro and in vivo fluorescence microphotographs of cells. (**C**) Scheme of the effect of VEGF associated with nanofibrils on cell signaling. Reprinted with permission from Elsevier, 2017 [159].

**Figure 4 polymers-12-00620-f004:**
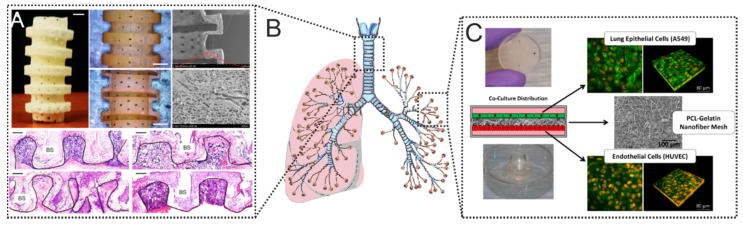
(**A**) Image of a tracheal implant (appearance and section). SEM micrograph of the outer wall and slice. Scalebar 2 mm. Histological Section 8 weeks after implantation. Reprinted with permission from Elsevier, 2015 [8]. (**B**) Schematics showing lungs, where 3D printed scaffolds shown in (**A**) can be incorporated, and which can be modeled by microfluidic devices depicted in (**C**). (**C**) Schematics of the co-culture system and images of assembled fluidic devices with cells including confocal laser scanning images of epithelial and endothelial cells co-cultured on a 75/25 (PCL/gelatin) fiber mesh. Reprinted from [173] with permission from Nature Publishing Group, © 2017.

**Figure 5 polymers-12-00620-f005:**
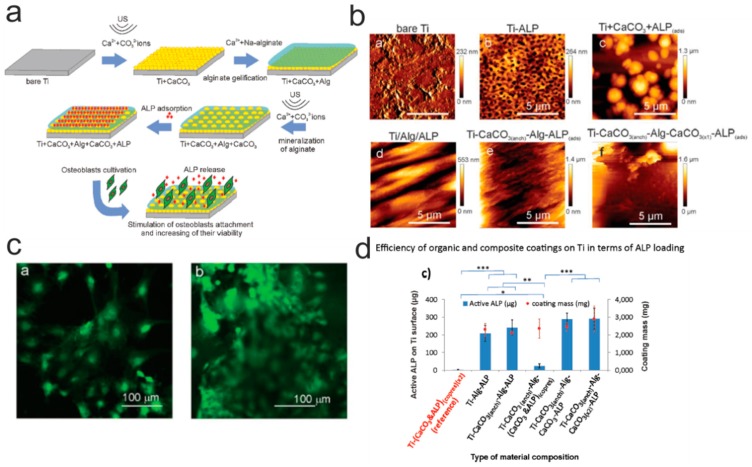
(**a**) Scheme depicting the creation of a composite material based on titanium and alginate gel. (**b**) Atomic force microscopy topography images of functionalized surfaces. (**c**) Fluorescence photographs showing osteoblast cells on the surface of a composite material. (**d**) Analysis of the amount of active enzyme by a different combination of components of the composite material. Reprinted with permission of Wiley, 2018 [228].

**Table 1 polymers-12-00620-t001:** Representative examples of polymer-and hybrid-based biomaterials with specific coating for interstitial, connective, vascular, nervous, visceral and musculoskeletal tissues.

Tissue Type	Tissue or Organ	Coating for Specific Tissue Type	Polymer as the Base of Coating	Functional Element of the Coating	Refs.
Interstitial and connective (Section 3)	Tendons (Section 3.1)	Collagen (hR COL and BAT COL), gelatin	PCL	Improving cell adhesion	[66,135,230]
		CaCO_3_ microparticles/PCL	PCL	Imitation of fibro-cartilaginous region and improving of cells adhesion	[138]
		Tricalcium phosphate nanoparticles/PCL	PCL	Imitation of fibro-cartilaginous region	[68]
		Collagen based	Silk	Improving cell adhesion	[231,232,233]
		Maleic anhydride/cellulose nanocrystal based	PLA	Improving connection between two components and cells	[136]
		Hap based	PLLA	Imitation of fibro-cartilaginous region	[137]
	Skin (Section 3.2)	Gelatin based	Chitosan	Macrophage activation, high hemostatic effect and no antigenicity.	[145]
		Collagen with immobilized epidermal growth factor (EGF)	PCL and PCL/Gelatin	Improving cells adhesion and stimulation of mitosis via regulation of Ca and pH inside cells	[146,147]
		Micelles with tannic acid	PCL/Collagen mix	Improving of angiogenesis and antimicrobial properties	[148]
		CaCO_3_ with tannic acid/PCL	PCL	Improving of angiogenesis and antimicrobial properties	[67]
		Platelet lysate	PCL/PLCL	Haemostatic plug stops bleeding	[234]
		PLGA based	VEGF and TGF-β3	Differentiation and angiogenesis of cells.	[144]
		PCL based	EGF + VEGF	Differentiation and angiogenesis of cells.	[152]
		Chitosan microparticles loaded with growth factors	Dextran-based hydrogel	Differentiation and angiogenesis of cells.	[153]
Vascular (Section 4)	Blood vessels (Section 4)	Fibrin glue with FGF-1 and heparin	ePTFE	Increasing capillarization, increased collagen content	[235]
		P15 peptide based	ePTFE	Smaller thicknesses of neointimal hyperplasia; enhanced endothelialization	[236]
		Growth factor -reduced Matrigel™-containing VEGF	ePTFE	Increased EC rate, increased myointimal hyperplasia	[237]
		Hyaluronic acid based	PCL	Higher bulk porosity and cell permeability	[238]
		VEGF based	PEA, PMA	Promoting vascularization in regenerative medicine applications	[159]
Nerve (Section 5)	Nerve (Section 5)	Laminin peptides	Chitosan	Enhancer of the attachment of neural cell, differentiation; outgrowth of neurites	[239]
		Platelet-rich plasma	PLA/GTNF	Healing improvement	[240]
		(γ-glycidoxypropyltrimethoxysilane (GPTMS)	Chitosan	Improving mechanical strength	[241]
		Poly-L-lysine, poly-L-ornithine, laminin, collagen	Poly (sialic acid)	Mechano compatibility; cell adhesive property	[242]
		Collagen type I; EDC/NHS activation (amino groups)	P(MMA-co-AA)	Promotion of cell attachment, spreading, and viability	[243]
		Polypyrrole (PPy)	PCL, PLA	Cell density, cell spread,	[244,245]
		RGD peptide based	PEG–heparin hybrid gel	Differentiation & propagation of NSCs; outgrowth of axon dendrites	[245,246]
		Laminin based	Collagen type I	Bridging peripheral nerve gaps	[245,247]
Visceral (Section 6)	Lung (Section 6.1)	Gelatin based	PCL PGA	Adherence of NCI-H292 cells; tuning mechanical properties of materials	[177,178,248]
		bFGF and AA2P based	PGA/PLA	Improving cells adhesion and stimulates cells proliferation and differentiation	[249]
	Kidney (Section 6.2)	Mg(OH)_2_ and acellular ECM	PLGA	Neutralization of the effects induced by degradation of PLGA products, suppression of inflammatory reactions	[186]
		l-DOPA and collagen IV	PCL	Improving cells adhesion and proliferation	[250]
	Liver (Section 6.3)	ECM-molecules	PLA	Microenvironment manipulation; changes of gene expression, protein contents, and cell adherence	[251]
			PCL		[252]
		HGF based	Gelatin	Controlled release for enhancement of the in vivo therapeutic effects	
		RGD and YIGSR based	PCL and PLA	Improved hepatocyte adhesion	[196]
Musculoskeletal (Section 7)	Cardiac muscles (Section 7.1)	bFGF based	PLGA	Neovascular formation	[253]
		Laminin based	PU	Improving cells adhesion	[254]
	Skeletal muscles (Section 7.2)	Collagen-chitosan based	PDMS	Improving cell adhesion	[255]
		Collagen-I based	Polyacrylamide	Improving cell adhesion	[256,257]
		Laminin and RGD based	PEG	Improving cells adhesion	[258,259]
		RGD based	Alginate	Improving cells adhesion	[75]
		Fibrin based	PLG	Anastomosis with the host vasculature and angiogenesis	[209]
		Agrin or laminin based	Alginate	Exhibited acetylcholine receptor (AChR) clustering	[210,211]
	Cartilage (Section 7.3)	Hyaluronic acid based	PLGA	Improving cell migration and proliferation	[45]
		Platelet-rich-plasma based	PGA	Homogeneous hyaline-like cartilage tissue repair.	[260]
		Fibrinogen based	PEG	Improving cells adhesion	[261]
		Fibrin based	PGA	Bio-sustainability of cell-based tissue regeneration (without in vitro cultivation)	[262,263]
		Hyaluronic acid based	Alginate	Improving cell migration and proliferation	[263,264]
		Tri-(calcium phosphate) based	PLGA	Improving osteochondral composition	[263,265]
	Bones (Section 7.4)	CaCO_3_/PHB & PHBV based	PHB and PHBV	Improving cells adhesion and ossification	[266]
		CaCO_3_ with loaded ALP	Alginate	Improving cells adhesion and ossification	[228]

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
