# Peer review of "Polymer- and Hybrid-Based Biomaterials for Interstitial, Connective, Vascular, Nerve, Visceral and Musculoskeletal Tissue Engineering"

_polymers, 2020, doi:10.3390/polym12030620_

Round 1

Reviewer 1 Report

This review summarizes polymer-based hybrid materials for tissue engineering, which is of interest to a broad readership. The review is generally well organized. I think it can be accepted after the authors carefully check through the expressions and grammar.

Author Response

Thank you. The English language in the article has been double checked and  improved. Some sentences have been reformulated.

Reviewer 2 Report

Manuscript requires proofreading of English.

Methodology for selecting literature sources should be described.

In several places the text is unclear, for example lane 36- "these two types of materials" -means which one?

The titles of the subsections are too general. For example "2. polymers- this is not a section about all polymers. "4. Vascular" or "5. Nervous" they are not good titles- These are shorthands.

In some parts of the manuscript, such as in "syntetic polymers" or "natural polymers" no mental continuity - no introduction to the chapter and information on subsequent types of polymers thrown in without continuity, the main idea.

Discussed by Decher (Science 1997) - there is no reference.

What this sentence means? ot English is not correct? "Composite LbL films containing inorganic nanoparticles"

Something wrong with text concerning figure 1. "Microphotographs show PCL fibers. (B);" It is not explained what is on part "B"

In two paragraphs concerning "vascular" lanes 254-263, there is no reference

Lane 273- it is stated "by the Salmerón-Sáncheza group. In that work..." there is no reference of Salmerón-Sánchez.

"Now, nanofiber frames consisting of natural and synthetic polymers are well studied" this sentence is not appropriate for scinece publication. What does it means "it is well studied"? do not use statements as this one.

In paragraph - lanes 305-307, there is no reference

RGD abreviation should be expalined

Lane 312 - "This decision increased the biological  activity of the growth factor [118]." Do not use "this decision" in scientific paper.

Lane 324 - another mistake in the citations- 2 times  reference 120.

Lane 339, there is mentioned "Park et al" but there is no indication of appropriate citation.

Lanes 370-375, there is no reference

The "Kidney" part is not well presented. There is no general statement what kind of materials are used or investigated. work of "Lih et al" is directly mentioned. We do not know why, what is the clue.

Lane 487, there is no reference to Muderrisogly et al. "165" is not appropriate.

There is no reference to the table in the text.

In the title of table it is mentioned "selected" but there is no information how the selection looked like. It was made on the basis of what criteria?

In "conclusions" part , lanes 505-520 there is no real conlusions from manusript, only generals statements...

Author Response

Manuscript requires proofreading of English.

The English language in the article has been double checked and  improved. Some sentences have been reformulated.

Methodology for selecting literature sources should be described.

The Introduction section has been expanded to better describe literature selection methods.

In several places the text is unclear, for example lane 36- "these two types of materials" -means which one?

The sentence has been reformulated.

The titles of the subsections are too general. For example "2. polymers- this is not a section about all polymers. "4. Vascular" or "5. Nervous" they are not good titles- These are shorthands.

Name of the sections were reformulated.

In some parts of the manuscript, such as in "syntetic polymers" or "natural polymers" no mental continuity - no introduction to the chapter and information on subsequent types of polymers thrown in without continuity, the main idea. Discussed by Decher (Science 1997) - there is no reference.

Thank you for pointing out this discrepancy. The following reference was added: Decher, G. Fuzzy Nanoassemblies: Toward Layered Polymeric Multicomposites. Science (80-. ). 1997, 277, 1232–1237.

What this sentence means? ot English is not correct? "Composite LbL films containing inorganic nanoparticles"

The sentence has been reformulated along with the pargraph.

Something wrong with text concerning figure 1. "Microphotographs show PCL fibers. (B);" It is not explained what is on part "B"

The describing has been reformulated.

In two paragraphs concerning "vascular" lanes 254-263, there is no reference

Thank you for point out to this discrepancy, to correct that we have added the following references:

Zhang, H.; Lui, K.O.; Zhou, B. Endocardial Cell Plasticity in Cardiac Development, Diseases and Regeneration. Circ. Res. 2018, 122, 774–789.

Kennedy, S.; Touyz, R.M. Anatomy and Pharmacology of Vessels. In Textbook of Vascular Medicine; Springer International Publishing: Cham, 2019; pp. 3–11.

The ESC Textbook of Vascular Biology; Krams, R., Bäck, M., Eds.; Oxford University Press, 2017; Vol. 1; ISBN 9780198755777.

Lane 273- it is stated "by the Salmerón-Sáncheza group. In that work..." there is no reference of Salmerón-Sánchez.

The last author in the article was accidentally used. Names of authors are corrected now.

"Now, nanofiber frames consisting of natural and synthetic polymers are well studied" this sentence is not appropriate for scinece publication. What does it means "it is well studied"? do not use statements as this one.

Indeed, this sentence sounded somewhat awkward. We have corrected that.

In paragraph - lanes 305-307, there is no reference

Thank you for pointing this out, the following references are added:

Zhao, Y.-Z.; Jiang, X.; Xiao, J.; Lin, Q.; Yu, W.-Z.; Tian, F.-R.; Mao, K.-L.; Yang, W.; Wong, H.L.; Lu, C.-T. Using NGF heparin-poloxamer thermosensitive hydrogels to enhance the nerve regeneration for spinal cord injury. Acta Biomater. 2016, 29, 71–80.

Burdick, J.A.; Ward, M.; Liang, E.; Young, M.J.; Langer, R. Stimulation of neurite outgrowth by neurotrophins delivered from degradable hydrogels. Biomaterials 2006, 27, 452–459.

Stanwick, J.C.; Baumann, M.D.; Shoichet, M.S. Enhanced neurotrophin-3 bioactivity and release from a nanoparticle-loaded composite hydrogel. J. Control. Release 2012, 160, 666–675.

Ghosh, B.; Wang, Z.; Nong, J.; Urban, M.W.; Zhang, Z.; Trovillion, V.A.; Wright, M.C.; Zhong, Y.; Lepore, A.C. Local BDNF Delivery to the Injured Cervical Spinal Cord using an Engineered Hydrogel Enhances Diaphragmatic Respiratory Function. J. Neurosci. 2018, 38, 5982–5995.

RGD abbreviation should be explained

As requested, the abbreviation is added (line 627.  All lines are indicated with tracking enabled)

Lane 312 - "This decision increased the biological  activity of the growth factor [118]." Do not use "this decision" in scientific paper.

Was changed to “This approach increased the biological activity of the growth factor [126].” Line 675

Lane 324 - another mistake in the citations- 2 times  reference 120.

Thank you, one of them was deleted.

Lane 339, there is mentioned "Park et al" but there is no indication of appropriate citation.

The reference to this article has been specified (Line 703)

Lanes 370-375, there is no reference

We thank the reviewer for pointing this out. The following references have been added:

Rayner, H.; Thomas, M.; Milford, D. Kidney Anatomy and Physiology. In Understanding Kidney Diseases; Springer International Publishing: Cham, 2016; pp. 1–10.

Lih, E.; Park, W.; Park, K.W.; Chun, S.Y.; Kim, H.; Joung, Y.K.; Kwon, T.G.; Hubbell, J.A.; Han, D.K. A Bioinspired Scaffold with Anti-Inflammatory Magnesium Hydroxide and Decellularized Extracellular Matrix for Renal Tissue Regeneration. ACS Cent. Sci. 2019, 5, 458–467.

The "Kidney" part is not well presented. There is no general statement what kind of materials are used or investigated. work of "Lih et al" is directly mentioned. We do not know why, what is the clue.

Section (6.2) of composite biomaterials applied to the kidneys has been expanded. Information on the use of hybrid hydrogels as carriers for cells is added.

Lane 487, there is no reference to Muderrisogly et al. "165" is not appropriate.

We thank to reviewer for pointing this out. We have gone through references providing necessary corrections. Line 914

There is no reference to the table in the text.

Yes, indeed, the reviewer is absolutely correct. To correct that we have added a new paragraph right after Figure 5 (in the new enumeration) describing various applications and summarizing the type of tissue and materials used for achieving targeted functionalities.  

In the title of table it is mentioned "selected" but there is no information how the selection looked like. It was made on the basis of what criteria?

Caption of the table was reformulated, where the term “Selected” was replaced with “Representative”. In selecting these examples no strict criteria were applied, the only goal was to include as many as possible relevant applications covering the identified tissues and materials.

In "conclusions" part , lanes 505-520 there is no real conlusions from manusript, only generals statements...

We thank the reviewer for pointing out this discrepancy. The conclusions and outlook have been rewritten to address this issue.
